# PaintSeg: Training-free Segmentation via Painting

**Xiang Li[1], Chung-Ching Lin[2], Yinpeng Chen[2], Zicheng Liu[2],**
**Jinglu Wang[2], Rita Singh[1], Bhiksha Raj[1,3]**
[1]CMU, [2]Microsoft, [3]MBZUAI
`xl6@andrew.cmu.edu`

## Abstract

The paper introduces PaintSeg, a new unsupervised method for segmenting objects without any training. We propose an adversarial masked contrastive painting (AMCP) process, which creates a contrast between the original image and a painted image in which a masked area is painted using off-the-shelf generative models. During the painting process, inpainting and outpainting are alternated, with the former masking the foreground and filling in the background, and the latter masking the background while recovering the missing part of the foreground object. Inpainting and outpainting, also referred to as I-step and O-step, allow our method to gradually advance the target segmentation mask toward the ground truth without supervision or training. PaintSeg can be configured to work with a variety of prompts, e.g. coarse masks, boxes, scribbles, and points. Our experimental results demonstrate that PaintSeg outperforms existing approaches in coarse mask-prompt, box-prompt, and point-prompt segmentation tasks, providing a training-free solution suitable for unsupervised segmentation. Code: `https://github.com/lxa9867/PaintSeg`.

## 1   Introduction

With deep learning advancements, significant progress has been made in the field of image generation and segmentation in recent years. A particular generative model, the denoising diffusion probabilistic model (DDPM), has demonstrated outstanding performance in a variety of generative tasks, such as image inpainting [49, 16, 34] and text-to-image synthesis [21, 20, 79]. Similar developments have occurred in the field of object segmentation, such as the strong zero-shot capability and excellent segmentation quality demonstrated by SAM [30].

Image generation and segmentation can be mutually beneficial. Segmentation has been shown to be a critical technique in improving the realism and stability of generative models by providing pixel-level guidance during the synthesis process [75, 29]. Interesting to note is the fact that the relationship between segmentation and generative models does not appear to be solely one-sided. Generative models learning to "paint" objects actually know where the painted object is. The emergence of unsupervised image segmentation methods utilizing generative adversarial networks (GANs) has produced a line of methods that can segment objects in images [4, 10, 5] using generative models. These methods work on the assumption that object appearance and location can be perturbed without compromising scene realism. By using the GAN architecture to discriminate between perturbed and real images, these methods can achieve effective object segmentation. Moreover, a follow-up work [61] develops an approach to leverage pre-trained GAN by identifying "segmenting" direction in the latent space to discriminate object shapes.

In this paper, we present PaintSeg, an approach for unsupervised image segmentation that leverages off-the-shelf generative models. Unlike previous methods [61, 4] that require training on top of these models, PaintSeg introduces a novel, training-free segmentation approach that relies on an adversarial masked contrastive painting (AMCP) process. The AMCP process creates a contrast

37th Conference on Neural Information Processing Systems (NeurIPS 2023).

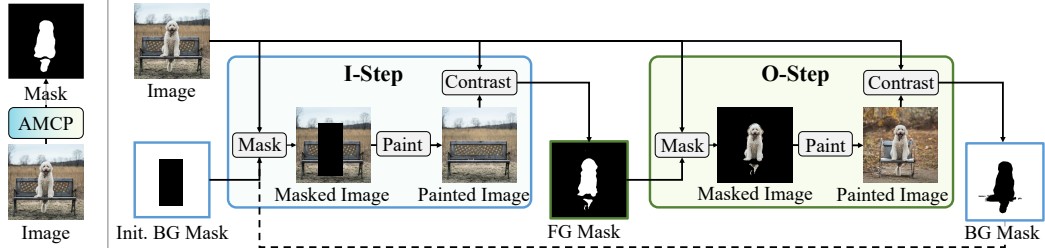

Figure 1: **Illustration of adversarial masked contrastive painting (AMCP).** Given an input image and an initial mask, AMCP leverages alternating I-step and O-step to gradually refine the segmentation mask until it converges to the ground truth. Both steps share the same mask, paint, and contrast operations. The updated mask in each step is achieved by binarizing the contrastive difference between the original and painted images.

between the original image and a painted image by alternating between inpainting and outpainting, with the former filling in the background and masking the foreground, and the latter retrieving the missing part of the object while masking the background and a portion of the foreground.

Both steps, as shown in Fig. 1, share the same operations while taking input from background and foreground masks, correspondingly. In the I-step, the object region is removed from the painted image, creating a significant contrast with the original image. Conversely, in the O-step, the background region exhibits a remarkable difference between the original and painted image. The foreground or background mask can be obtained by binarizing the contrastive difference in each step.

Although either I-step or O-step is capable of discriminating objects, the single-step method is less robust. The I-step involves segmenting objects based on background consistency without taking into account object information. As a result, the segmentation may be imperfect if the object part resembles the background. Similarly, in the O-step, only the object shape prior is utilized, resulting in a lack of background knowledge. This problem is addressed by introducing adversarial mask updating, in which I-steps and O-steps are alternated. During I-step, we only shrink the object mask to cut off background false positives, while during O-step, we expand it to link up foreground false negatives. Thereby, even if errors occur during the iteration of AMCP, they will be corrected in the next step without degradation. With the adversarial mask updating, the target mask can be gradually advanced to the ground truth.

With the robustness of AMCP, PaintSeg can deal with inaccurate initial masks and adapt to various visual prompts, such as coarse masks, bounding boxes, scribbles, and points. Compared to the recently published successes in image object segmentation study, our main contributions are as follows:

- We propose PaintSeg, a training-free approach to segmenting image objects based on heterogeneous visual cues. The method provides a direct bridge between generative models and segmentation.

- We introduce adversarial masked contrastive painting (AMCP), consisting of alternating I-step and O-step, to robustly segment objects.

- We conduct extensive experiments for analysis and comparisons on seven different image segmentation datasets, the results of which show the superiority and generalization ability of our methods.

## 2 Related Works

### 2.1 Unsupervised Image Segmentation

Unsupervised methods for image segmentation are extensively investigated with the advancements in self-supervised. DINO [8] provides a self-supervised approach to explicitly bring out underlying semantic segmentation of images using a Vision Transformer (ViT) [17]. Based on DINO, LOST [56], Deep Spectral Methods [44] and TokenCut [66] leverage self-supervised ViT features and propose to

segment objects using NCut [52]. Subsequently, [57, 55] introduce a second-stage training approach to further improve the segmentation quality. Found [57] incorporates background similarity as an additional refinement factor, while SelfMask [55] utilizes an ensemble of features [7, 8, 12] to enhance image representation. CutLER [65] enables multiple objects discovery capability by iteratively cutting objects with NCut and introduces a more powerful second-stage training. FreeSOLO [64] generates coarse masks with correlation maps that are then ranked and filtered by a "maskness" score. Another line of unsupervised methods learns to generate a realistic image by combining a foreground, a background and a mask [5, 71, 70, 59, 32, 18, 25] and then the object segmentor can be obtained as a byproduct.

## 2.2 Prompt-guided Segmentation

Prompt-guided segmentation aims to segment objects assigned by prompts, e.g., mask, box, scribble and point. Semi-supervised video object segmentation (VOS) [1, 68, 38], aiming at segmenting object masks across frames given the first frame mask, is a typical mask-prompt task. The mainstream of VOS methods [72, 73] constructs pixel-level correspondence and propagates masks by exploring matches among adjacent frames. Interactive segmentation (IS) [78, 40, 58, 23] is another line of prompt-guided segmentation. IS permits users to leverage scribbles and points to assign target objects and segment them. In addition, an interactive correction is also featured by IS which introduces additional prompts to correct misclassified regions. MIS [33] is a recent work tackling unsupervised IS and proposes a multi-granularity region proposal generation to refine the mask. SAM [30] is a recently introduced zero-shot method for prompt-based segmentation which introduces a large-scale dataset and a strategy to mitigate the ambiguity of prompt. Beyond visual prompts, objects can also be referred by natural language or acoustic prompts. Referring image segmentation (RIS) [28, 74] and referring video object segmentation (R-VOS) [11, 76, 15, 35, 37] aims to segment objects in image/video referred by linguistic expressions. Audiovisual segmentation [77, 36] aims to segment sound sources in the given audiovisual clip.

## 2.3 Conditional Image Generation

Conditional image generation refers to the process of generating images based on specific conditions or constraints. In most instances, the condition can be based on class labels, partial images, semantic masks, etc. Cascaded Diffusion Models [26] uses ImageNet class labels as a condition to generate high-resolution images with a two-stage pipeline of multiple diffusion models. [51] guides diffusion models to produce novel images from low-density regions of the data manifold. Apart from these, CLIP [48] has been widely used in guiding image generation in GANs with text prompts [21, 20, 79]. For diffusion models, Semantic Diffusion Guidance [41] investigates a unified framework for diffusion-based image generation with language, image, or multi-modal conditions. Dhariwal et al. [14] apply an ablated diffusion model to use the gradients of a classifier to guide the diffusion with a trade-off between diversity and fidelity. Additionally, Ho et al.[27] introduce classifier-free guidance in conditional diffusion models by mixing the score estimates of a conditional diffusion model and a jointly trained unconditional diffusion model.

# 3 Problem Definition

We tackle the unsupervised prompt-guided image object segmentation task, which aims to predict the object mask $M \in \{0, 1\}^{1 \times H \times W}$ in an image $I \in \mathbb{R}^{3 \times H \times W}$ given a visual prompt $P \in \{0, 1\}^{1 \times H \times W}$. The visual prompt can have a format of a point, a scribble, a bounding box or a coarse mask of the target object $P \in \{P_{point}, P_{scrib}, P_{box}, P_{mask}\}$. Following the convention, we assume the ground-truth object mask $M$ must have an overlap with the visual prompt $P \cap M \neq \emptyset$.

# 4 Adversarial Masked Contrastive Painting

PaintSeg leverages adversarial masked contrastive painting (AMCP) to gradually refine the initial prompt $P$ to the object mask $M$. The AMCP approach is composed of alternating I-steps and O-steps, as illustrated in Figure 1. During each step, a region of the image is masked out based on the previous iteration's mask, and the masked region is then repainted and compared to the original image to refine the mask prediction. To improve the segmentation's robustness, PaintSeg introduces adversarial

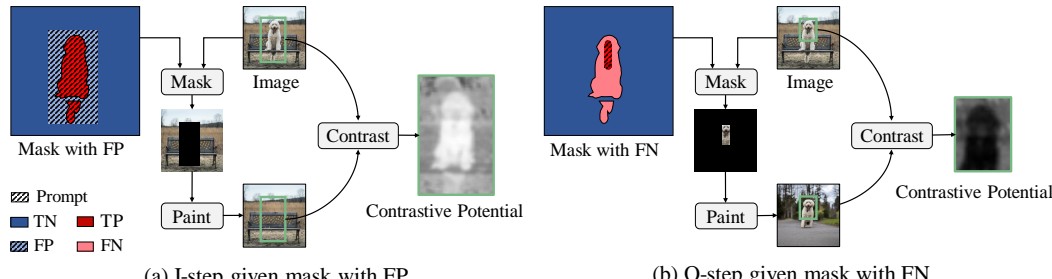

(a) I-step given mask with FP        (b) O-step given mask with FN

Figure 2: **Illustration of I-step and O-step with initial prompts.** (a) We show an I-step with a box prompt as the initial mask, where the object region has a significant difference between the original and painted images. (b) We show an O-step with a scribble prompt as the initial mask, where the object region has a small difference between the original and painted images.

mask updating, which helps to ensure that the mask accurately reflects the object's boundaries. The I-step is used to shrink the object mask by leveraging background consistency, thereby eliminating false-positive regions. On the other hand, the O-step expands the object mask by utilizing object shape consistency to link up false-negative foreground regions.

## 4.1 Contrastive Painting

We first discuss the rationality of segmenting objects by contrasting painted and original images. Given a visual prompt $P$, the relation between the prompted area and object mask can be categorized into three types: background false positive (Fig. 2 (a)), foreground false negative (Fig. 2 (b)) and a hybrid of both. We tackle the prompt-guided segmentation by separately addressing the background false positive and foreground false negative with I-step and O-step respectively.

We discuss the painted content with different mask situations. To avoid ambiguity, we first denote the generative model taking background and foreground as conditions as inpainting model $\phi(\cdot)$ outpainting model $\psi(\cdot)$ respectively. We consider the prompted area as the initial mask $M_0 = P$. When the initial mask has false positives, i.e., $M \subset M_0$, as shown in Fig. 2 (a), the inpainted content tends to complete the background based on the background consistency. In this way, the inpainted pixels inside the object will have a significant difference compared to the original image. In contrast, when the initial mask has false negatives, i.e., $M_0 \subset M$, as shown in Fig. 2 (b), the outpainted content tends to complete the partial object leading to a low difference with the original image inside object region. We notice that I-step can address background false-positive and O-step can address foreground false-negative. By alternating conducting I-step and O-step, we can leverage both foreground and background consistency and address more complicated cases.

## 4.2 Contrastive Potential

Given the a image $I$ and a mask $M_t$, we define a contrastive potential $\Phi$ to measure the region relations, which contains three terms

$$\Phi = \lambda_{paint}\Phi_{paint} + \lambda_{color}\Phi_{color} + \lambda_{prompt}\Phi_{prompt}. \tag{1}$$

We introduce a box region $B$ that encloses the foreground and define the $\Phi_{paint}$ term as the distance between the painted and the original image. Specifically, $\Phi_{paint} = B \circ |\mathcal{E}(I) - \mathcal{E}(I_{paint})|_2$, where $\mathcal{E} : \mathbb{R}^{3 \times H \times W} \to \mathbb{R}^{C \times H \times W}$ is a function that projects the image to a high-dimensional space. $\circ$ denotes the Hadamard product.

The $\Phi_{color}$ term measures the pixel-level color similarity inside and outside the mask $M_t$. To compute it, we use the output of the conditional random field algorithm [31] and define $\Phi_{color} = \mathcal{C}(M_t \circ B, I \circ B)$, where $\mathcal{C}$ is a function that takes a mask and an image as inputs and outputs the probability of whether a pixel should belong to the masked region.

To further incorporate prompt information, we introduce prompt priors $\Phi_{prompt}$ to the contrastive potential for box, scribble, and point prompts. Let us denote $[x_l, y_l]$ as the coordinates of the $l$-th

point in the prompt area. The prompt prior is defined as follows:

$$\Phi_{prompt}[i,j] = \max_l \mathcal{G}(x_l, y_l)[i,j], \tag{2}$$

where $\mathcal{G}$ is a two-dimensional Gaussian function, and $\mathcal{G}[i,j] = \exp(\frac{(i-x_l)^2}{\sigma_x^2} + \frac{(j-y_l)^2}{\sigma_y^2})$. Specifically, for the box prompt, we only take the center point of the box into account. The prompt priors are designed to leverage the positional information of the prompts to better locate the target object. By taking the maximum value of the Gaussian function over all points in the prompt area, $\Phi_{prompt}$ captures the overall strength of the prompt signal.

## 4.3 Adversarial I/O-Step

As shown in Fig. 1, I-step and O-step share the same mask, paint, and contrast processes while the input mask in I-step is the background mask and, in O-step, the foreground mask. Given the original image $I$ and an input mask $M_t$ from the $t$-th step of AMCP (assuming $M_t$ is a background mask thus $t+1$-th step is an I-step), we first filter out the masked region by $I \circ M_t$ and then paint the image $I_{paint} = \phi(I \circ M_t)$. As discussed in Section 4.1, the foreground region will have a significant difference between painted and original images. We obtain the updated mask $M_{t+1}$ by k-means clustering $\mathcal{K}(\cdot)$ over the contrastive potential $\Phi$. Let us denote $\mu_k$ and $S_k \in \{0,1\}^{H \times W}$ as the average value of all samples in the $k$-th cluster and its corresponding identity map ($S_k[i,j] = 1$ if pixel $[i,j]$ belongs to center $k$ else 0). The updated mask can be found by

$$M_{t+1} = S_{k^*}, \ k^* = \arg\max_k \mu_k. \tag{3}$$

The updated mask $M_{t+1}$ is a foreground mask thus the next step will be an O-step. Similarly, we paint the image by $I_{paint} = \psi(I \circ M_{t+1})$. Here, the difference in background area will have a significant difference between painted and original images. Thereby, the updated mask $M_{t+2}$ from O-step can be computed using the same rule as Eq. (3) which leads to a background mask. By updating the mask by Eq. (3), we notice that when the input mask $M_t$ is a background mask, then the output mask will be a foreground mask and vice versa. Thereby, the alternating I-step and O-step can be automatically achieved.

As discussed in Section 4.1, I-step is advantageous for reducing false positives in the background, whereas O-step is beneficial for reducing false negatives in the foreground. Specifically, the updated mask is configured to only cut off pixels in the I-step, and to only link up pixels in the O-step. Let $M_t^+$ and $M_t^-$ as the dilated and eroded masks of $M_t$. We constrain to only update the regions near the foreground-background boundary. In this way, the updating rule for AMCP can be rewritten as

$$M_{t+1} = \begin{cases} S_{k^*} \circ \Delta^- + \bar{M}_t \circ (1 - \Delta^-), & \text{I-step} \\ S_{k^*} \circ \Delta^+ + \bar{M}_t \circ (1 - \Delta^+), & \text{O-step} \end{cases}, \ k^* = \arg\max_k \mu_k \tag{4}$$

where $\Delta^- = \bar{M}_t - \bar{M}_t^-$ and $\Delta^+ = M_t^+ - M_t$ are the inner and outer neighbors of $M_t$. Through the adversarial alternation of I-steps and O-steps, AMCP can handle more complex cases involving both false positives and false negatives. Due to the randomness inherent in generative painting, we paint the image $N$ times in each step, and use the averaged mask as an output.

## 4.4 Discussion

In this section, we introduce the mathematical formulation of AMCP. Mathematically, an image can be represented as a masked combination of a foreground image $I_F$ and a background image $I_B$

$$I = I_F \circ M + I_B \circ \bar{M}, \ M \in \{0,1\}^{H \times W \times 1}. \tag{5}$$

$M$ is a foreground mask. $\bar{M} = 1 - M$. An inpainting model $\phi[\cdot]$ is defined to generate pixels inside the mask given the pixels outside the mask as a condition. Similarly, an outpainting model $\psi[\cdot]$ predicts pixels outside the mask given the pixels inside the mask as a condition. In our method, we aim to find a $M$ that maximizes

$$\arg\max_M \underbrace{\left\| I \circ \Delta^- - \phi(I \circ \bar{M}) \circ \Delta^- \right\|_d}_{\text{I-Step}} + \underbrace{\left\| I \circ \Delta^+ - \psi[I \circ M] \circ \Delta^+ \right\|_d}_{\text{O-Step}} \tag{6}$$

| Method | Training | DUTS-TE [62] | ECSSD[54] |
|---|---|---|---|
| — *Compared to methods with training* — | | | |
| SelfMask [55] CVPRW22 | ✓ | 62.6 | 78.1 |
| SelfMask [55] CVPRW22 + BS [2] | ✓ | 66.0 | **81.8** |
| FOUND [57] CVPR23 | ✓ | 63.7 | 79.3 |
| FOUND [57] CVPR23 + BS [2] | ✓ | 66.3 | 80.5 |
| **PaintSeg** | | **67.0** | 80.6 |
| — *Compared to methods without training* — | | | |
| Melas-Kyriazi et al. [43] ICLR22 | | 52.8 | 71.3 |
| LOST [56] BMVC21 | | 51.8 | 65.4 |
| LOST [56] BMVC21 + BS [2] | | 57.2 | 72.3 |
| DSS [45] CVPR22 | | 51.4 | 73.3 |
| TokenCut [66] CVPR22 | | 57.6 | 71.2 |
| TokenCut [66] CVPR22 + BS [2] | | 62.4 | 77.2 |
| SelfMask† [55] CVPRW22 | | 46.6 | 64.6 |
| FOUND† [57] CVPR23 | | - | 71.7 |
| **PaintSeg** | | **67.0** | **80.6** |

Table 1: **Qantitative results of coarse mask-prompted segmentation on DUTS-TE and ECSSD.** PaintSeg utilizes the coarse mask generated by unsupervised TokenCut [66] as prompt. BS denotes the application of the post-processing bilateral solver on the generated masks and the column 'Learning' specifies which methods have a training step. The best result per section is highlighted in **bold**. The second best result for each section is underlined. † indicates the first-stage pseudo mask obtained without training.

The first term aims to maximize the difference between the original image $I$ and the inpainted image $\phi(I \circ \bar{M})$ in the inner neighbor $\Delta^-$ which corresponds to the I-step in AMCP. The second term aims to maximize the difference between the original image $I$ and the outpainted image $\psi[I \circ M]$ in the outer neighbor $\Delta^+$ corresponding to the O-step.

In each step, our mask, paint, and contrast operations can be considered as an expectation-maximization-like (EM-like) process with the latent variable of $I_{paint}$ to maximize Eq. (6). On one hand, the $I_{paint}$ is estimated by the mask and paint operations where the conditional probability $p(I_{paint}|I, M)$ is characterized by the generative painting models (expectation step). On the other hand, the predicted mask $M$ can be updated by maximizing the contrastive potential $\Phi$ (maximization step). Since the EM algorithm is sensitive to the initial value, solely updating with I-step or O-step cannot achieve robust performance. With the alternating I-step and O-step, we introduce an adversarial updating process which leads to a more robust mask estimation.

## 5   Experiment

### 5.1   Datasets

For mask-prompt segmentation, we evaluate on DUTS-TE [63] and ECSSD [53]. DUTS-TE contains 5,019 images selected from the SUN dataset [67] and ImageNet test set [13]. ECSSD [53] contains 1,000 images that were selected to represent complex scenes. For box-prompt segmentation, we evaluate on PASCAL VOC [19] val set and COCO [39] MVAL datasets. COCO MVal contains 800 object instances from the validation set with 10 images from each of the 80 categories. For point-prompt segmentation, we use three datasets including GrabCut [50] which contains 50 images and corresponding segmentation masks that delineate a foreground object; Berkeley [42] which contains 96 images with 100 instances with more difficulty than GrabCut and DAVIS [47] which is a video dataset and 10% of the annotated frames are randomly selected, yielding 345 images that are used in the evaluation

### 5.2   Experimental Setup

**Evaluation metrics.**   In accordance with previous methods [30, 66], we evaluate segmentation quality using intersection over union (IoU).

**Implementation details.**   We leverage the inpainting models trained with latent-diffusion pipeline [49] as our $\phi$ and $\psi$. We set the diffusion iterations to 50. We leverage DINO [8] pretrained VIT-S/8

| Method | Training | Supervision | GrabCut | Berkeley | DAVIS |
|---|---|---|---|---|---|
| — *Compared to methods with training* — | | | | | |
| DIOS[69] $_{\mathrm{CVPR16}}$ | ✓ | ✓ | 64.0 | 66.0 | 57.8 |
| RITM [58] $_{\mathrm{ICIP22}}$ | ✓ | ✓ | 81.0 | **77.7** | 66.0 |
| MIS [33] $_{\mathrm{arXiv23}}$ | ✓ | | 76.2 | 63.2 | 53.3 |
| **PaintSeg** | | | **84.4** | 70.0 | **69.4** |
| — *Compared to methods without training* — | | | | | |
| Random Walk [22] $_{\mathrm{TPAMI06}}$ | | | 25.7 | 26.2 | <20 |
| GrowCut [60] $_{\mathrm{GraphiCon05}}$ | | | 26.7 | 26.2 | - |
| GraphCut [6] $_{\mathrm{ICCV01}}$ | | | 41.8 | 33.9 | <20 |
| **PaintSeg** | | | **84.4** | **70.0** | **69.4** |

Table 2: **Qantitative comparison of point-prompted segmentation on GrabCut, Berkeley, and DAVIS.** The point prompt is given as the centroid of each object.

| Method | Training | Supervision | PASCAL VOC | MVal |
|---|---|---|---|---|
| — *Compared to methods with training* — | | | | |
| Mask-RCNN [24] $_{\mathrm{ICCV17}}$ | ✓ | ✓ | **73.2** | **79.4** |
| CutLER [65] $_{\mathrm{CVPR23}}$ | ✓ | | 63.5 | 74.8 |
| **PaintSeg** | | | 59.7 | 69.6 |
| — *Compared to methods without training* — | | | | |
| TokenCut [66] $_{\mathrm{CVPR22}}$ | | | 30.2 | 34.7 |
| **PaintSeg** | | | **59.7** | **69.6** |

Table 3: **Qantitative comparison of box-prompted segmentation on PASCAL VOC and COCO MVal.**

[17] as our $\mathcal{E}$. We use [31] as our $\mathcal{C}(\cdot)$ to calculate $\Phi_{color}$. If no specification, for all experiments, the masked contrastive painting starts from the I-step and updates for 5 steps. We set the number of cluster centers to 3 in the first three steps for point, box and scribble prompts otherwise 2. We set $\lambda_{paint} = 0.8$, $\lambda_{color} = 0.2$ and $\lambda_{prompt} = 0.2$ if in I-step and $\lambda_{prompt} = -0.2$ if in O-step. We average N=5 painted images to obtain the updated mask for each step. The $\sigma_x$ and $\sigma_y$ are set to $\frac{1}{10}$ of the width and height of the bounding box of the current stage mask respectively. $\Delta^+$ and $\Delta^-$ are the neighbors 32 pixels outside and inside the object boundary. We leverage dilation and erosion to filter out sparse points for each iteration. The kernel size is set to 5. For the mask and box prompts, we set the prompt as the initial mask. For the point and scribble prompts, we set the entire image as the initial masked region. The images are padded to $512 \times 512$ to fit the generative inpainting model.

### 5.3 Main Results

**Coarse mask prompt.** Since the usage of the ground-truth coarse mask as a prompt is rare, we evaluate PaintSeg on two unsupervised salient object detection benchmarks and leverage the coarse mask generated from TokenCut [66] as our prompt. As shown in Table 1, PaintSeg achieves encouraging performance that is even comparable with training-based methods. Under the training-free setting, PaintSeg significantly outperforms previous methods by a margin of 4.6 IoU on DUTS-TE and 3.4 IoU on ECSSD. We attribute the performance improvement to the error correction capability of PaintSeg. With alternating between I-step and O-step, the proposed PanintSeg can handle noisy prompts effectively. The robustness of PaintSeg will be discussed in more detail in Section 5.4.

**Point prompt.** As shown in Table 2, we compare our method with state-of-the-art point prompt segmentation approaches. PaintSeg consistently outperforms the training-free methods. Even compared to training-based methods with ground truth supervision, PaintSeg still achieves the best performance on GrabCut and DAVIS datasets. MIS [33] is an unsupervised approach equipped with second-stage training. We notice that our method can significantly outperform it in terms of IoU, with improvements of 8.2, 6.8, and 16.1 on GrabCut, Berkeley, and DAVIS correspondingly.

**Box prompt.** Since there is no unsupervised box-prompted segmentation that can be directly compared, we compare the proposed method with several baselines including TokenCut [66], CutLER [65] and MaskRCNN [24]. We first cut off the ground truth box region and then run the baselines. As shown in Table 3, when compared with training-based Mask-RCNN and CutLER, PaintSeg shows suboptimal performance, which can be explained by the lack of training to handle complex

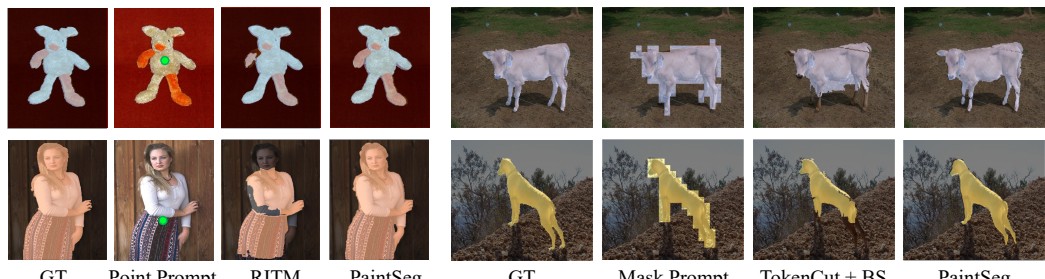

| GT | Point Prompt | RITM | PaintSeg | GT | Mask Prompt | TokenCut + BS | PaintSeg |

Figure 3: **Qualitative results of baselines and our PaintSeg with point and mask prompts.** Green point denotes the point prompt. The mask prompt is generated by unsupervised TokenCut [66]. BS represents the bilateral solver [3]. We compare with RITM [58] and TokenCut [66].

| I-step | O-step | AC | IoU |
|--------|--------|-----|------|
| ✓ | | | 78.4 |
| | ✓ | | 77.9 |
| ✓ | ✓ | | 79.5 |
| ✓ | ✓ | ✓ | 80.6 |

| Prompt | 0% Noise | 15% Noise | 30% Noise |
|--------|----------|-----------|-----------|
| Point | 60.8 | 60.4 | 58.9 |
| Scribble | 64.3 | 63.7 | 62.7 |
| Box | 71.0 | 70.7 | 70.1 |
| Coarse Mask | 80.6 | 80.0 | 79.3 |

Table 4: **Module effectiveness in AMCP**. AC: adversarial constraint for mask updating.

Table 5: **Prompt robustness.** We add random noise to the prompt to evaluate the robustness of AMCP. The noise scale is determined by half the length of the object box diagonal.

scenarios. However, as MaskRCNN is trained on 80 COCO object categories, the "unseen" gap remains substantial. PaintSeg provides an alternative solution that is not reliant on training, thus making it more general and capable of handling new categories of objects. When compared with unsupervised approaches, our method eclipses TokenCut by a large margin on both PASCAL VOC and COCO MVal datasets.

**Qualitative results.** We visualize the qualitative results with point and coarse mask prompt in Fig. 3. Our visualization depicts comparably reliable results. Comparatively, PaintSeg segments a relatively complete object, while baselines miss some parts of it.

### 5.4 Analyses

**Module effectiveness in AMCP.** We step by step add proposed modules in AMCP to validate the effectiveness. As shown in Table 4, we report the results on ECSSD with coarse-mask prompts. We observe that the missing of either step impacts the performance, as evidenced by the significant drop in IoU (compared to alternating I-step and O-step). With the adversarial mask updating constraint, AMCP achieves the best performance of 80.6 IoU.

**Robustness of AMCP with different prompts.** In Table 5, we add noise to the initial prompt by randomly shifting the position to investigate the robustness of AMCP. The scale of random noise is determined, w.r.t., half the length of the diagonal of the ground-truth bounding box. We observe that AMCP remains robust and only shows a slight performance drop with a noise rate of less than 30%. The robust capability can be attributed to 1) the alternating I-step and O-step to leverage both background and object shape consistency, and 2) the adversarial mask updating to tackle the background false-positives and foreground false-negatives.

**Design choices in AMCP.** We conduct experiments to ablate the design choices in AMCP and their impacts on the segmentation performance. We first study the effect of cluster center numbers for quantizing contrastive potential. With a larger cluster center, AMCP will ignore more ambiguous regions. As shown in Table 6a, we notice a cluster center of 2 achieves the best performance for mask prompt. After that, we ablate on the AMCP step number in Table 6b. The segmentation performance keeps increasing until reaching a step number of 5. In this way, we choose 5 as our step number. As we leverage the diffusion-based generative model, we ablate the iterations for the diffusion process as

| K | 2 | 3 | 4 | | $T$ | 3 | 4 | 5 | 6 | | Iter | 10 | 30 | 50 | | Rate | 0.9 | 1.0 | 1.1 | 1.2 |
|---|---|---|---|---|---|---|---|---|---|---|---|---|---|---|---|---|---|---|---|---|
| IoU | **80.6** | 72.3 | 61.5 | | IoU | 78.4 | 79.1 | **80.6** | 80.5 | | IoU | 77.3 | 78.7 | **80.6** | | IoU | 69.2 | 72.4 | **80.6** | 80.0 |

(a) **Cluster center**.  (b) **Step number.**.  (c) **Iter. for painting**.  (d) **Box size for contrasting**.

Table 6: **Design choices for AMCP.** We report the performance with the coarse-mask prompt on ECSSD. (a) We ablate the cluster center when contrasting. (b) We ablate the step number for AMCP. (c) We ablate the diffusion iteration for generative painting. (d) We ablate on the cropped box size when contrasting. The rate denotes the proportion of cropped mask and the box of the current stage object mask.

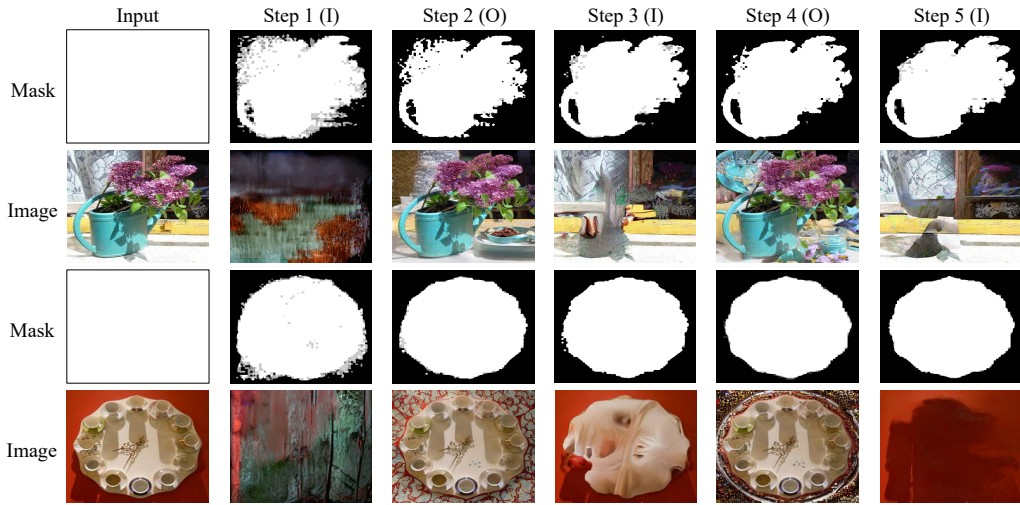

Figure 4: **Iterative process of AMCP with box prompt.** We inverse the outputted background mask in I-step for better comparison. We only visualize the box prompted area.

it can impact the image quality. As expected, Table 6c demonstrates that a larger iteration number can reach a better performance. To filter out irrelevant background regions, we crop a box region wrapping the given object mask to contrast images. We ablate the box size in Table 6d. We notice that a box slightly larger than the bounding box to the given mask can achieve the best performance. An explanation for this could be that a box tightly enclosing an object will result in a high proportion of object region, which may dominate the features and lead to ambiguity. Properly introducing background can make the extracted features more discriminative and easier for clustering.

**Visualization of mask updating.** To better illustrate the iterative process of AMCP, as shown in Fig. 4, we visualize the averaged mask output (among $N$ painted images in each step) for each step with a box prompt. As the given mask only contains background false positives, I-step plays a major role to cut false-positive backgrounds in AMCP. The mask shrink can also be observed after the O-step which is due to the binarization of the averaged mask from the I-step instead of contrastive painting. We observe that the updated masks are gradually closer to the mask of the target object with AMCP.

## 6 Conclusion

To conclude, PaintSeg bridges the gap between generative models and segmentation. It is designed to provide a robust and training-free approach to unsupervised image object segmentation. With the proposed adversarial masked contrastive painting (AMCP) process, PaintSeg creates a contrast between the original image and the painted image by alternately applying I-steps (inpainting) and O-steps (outpainting). The alternating I-step and O-step gradually improve the accuracy of the object mask by leveraging consistency in the background and the shape of the object. The competitiveness of our method on seven different image segmentation datasets suggests that PaintSeg can deal with inaccurate initial masks and adapt to various visual prompts, such as coarse masks, bounding boxes,

scribbles, and points. An extensive ablation analysis indicates a number of key factors and advantages of the proposed model, including its design choices and generalizability.

**Limitation.** In spite of PaintSeg's high performance for training-free image segmentation with heterogeneous visual prompts, it does not possess object discovery capabilities and therefore cannot automatically recognize instance-level masks in an image. Developing discovery capability can be achieved by conducting second-stage training on the segmentation results generated by PaintSeg, which is our future research focus.

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

# A More Comparison with Mask-RCNN

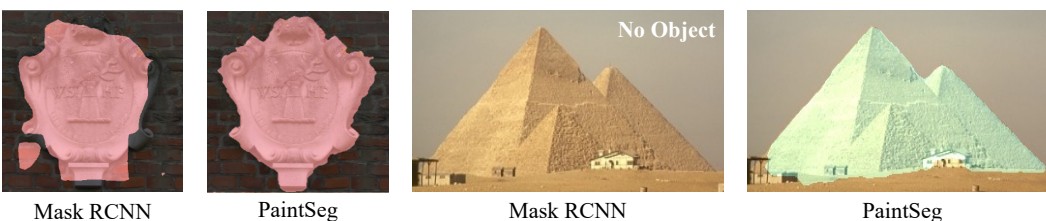

Figure A: **Comparison with Mask RCNN with objects beyond 80 COCO categories.**

We present more results compared with supervised Mask RCNN [24]. As shown in Fig. A, we compare box-prompted segmentation with Mask RCNN on objects beyond 80 COCO categories. In the shown examples, we observe that Mask RCNN has difficulty segmenting the correct shape of the object. Instead, PaintSeg provides more accurate object segmentation. As Mask RCNN is only trained on 80 COCO object categories, there is still a substantial gap between the seen and the unseen. In contrast, PaintSeg is a solution that does not require training, which makes it more general and capable of handling new object categories.

# B More Ablation Experiments

In this section, we provide additional ablation studies to illustrate the design choices of PaintSeg.

| N | 1 | 2 | 3 | 4 | 5 | 6 |
|---|---|---|---|---|---|---|
| IoU | 78.8 | 79.2 | 79.6 | 80.1 | 80.6 | 80.8 |

Table A: **Ablation study on the painted image number $N$ for each step.**

## B.1 Sampling Number for Each Step

We average $N$ painted images in each step to obtain the final mask prediction due to the randomness of the generative painting model. We present an ablation study to illustrate the impact of the number of painted images in each step. As shown in Table A, we report the performance on the ECSSD [54] dataset with coarse mask prompt from TokenCut [66]. We notice that the performance gradually improved with more painted images averaged in each step. As there is no significant difference in performance between five or six painted images used, we set the number of painted images to five in the PaintSeg process.

| DINO [9] VIT-S/8 | DINO-V2 [46] VIS-S/14 |
|---|---|
| 80.6 | 80.0 |

Table B: **Ablation study on image projector $\mathcal{E}$ used in AMCP.**

## B.2 Image Projector

We conduct an ablation study on image projector $\mathcal{E}$ as illustrated in Table B. We compare the widely used DINO [9] VIT-S/8 and the latest DINO [46] VIS-S/14. The results demonstrate that DINO with a small patch size achieves better performance. It follows that we consider a smaller patch size since PaintSeg requires fine-grained visual information. A larger patch size will blur the object boundary, resulting in a performance drop.

## C   More Potential Application

In this section, we discuss more potential applications of PaintSeg beyond prompt-guided object segmentation.

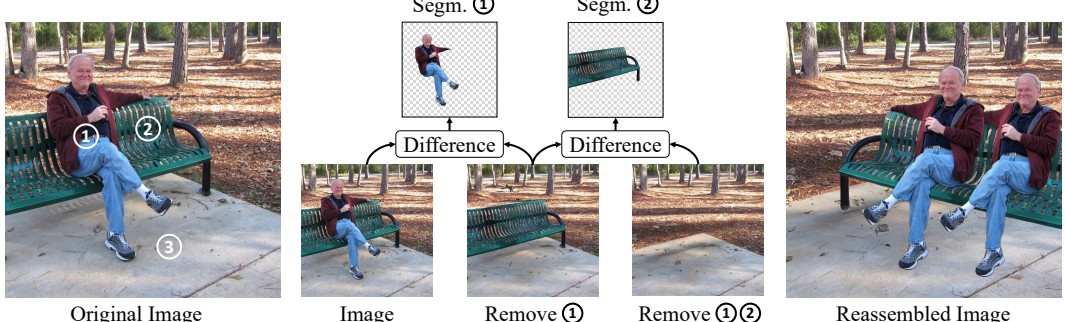

Figure B: **Potential application in image edition and amodal segmentation.** PaintSeg can step-by-step remove objects in the image by using the painted image in I-step. With the segmented object and painted image without objects, we can freely assemble them into a new image. Further, PaintSeg supports amodal segmentation, with the painting capability enabling segmentation of the occluded areas.

### C.1   Image Edition

In the I-step of AMCP, the painted image will remove the target object while keeping all other contents in the image. In this way, with the segmented objects and an image without target objects, we can reassemble them into a new image as shown in Fig. B.

### C.2   Amodal Segmentation

As shown in Fig. B, PaintSeg can layer-by-layer segment objects. By using the painted image in I-step as the input to the next iteration, PaintSeg can attach the amodal capability. We notice that the bench is occluded by the men in Fig. B. With the PaintSeg, the full shape of the bench can be segmented.

## D   More Discussion about PaintSeg

### D.1   Point Prompt

As depicted in Fig 2, the object mask can be derived by contrasting in/outpainted images with the original ones. In the I-step, there is a substantial difference (Fig 2 (a)) in the object area. In contrast, in the O-step, the object area has a small difference (Fig 2 (b)).

To incorporate the point information, we introduce a prompt term $\Phi_{prompt}$ in the contrastive potential to enhance the difference based on the point location. Specifically, $\Phi_{prompt}[i,j] = \mathcal{G}(x,y)[i,j]$ (Equ 2) represents a gaussian offset. Here, $(x,y)$ refers to the point prompt, serving as the mean, while $\frac{1}{10}$ of the bounding box's height and width determine the variance (Line 159). During the I-step, we add $\Phi_{prompt}$ to the contrastive potential to amplify the difference in the prompted area. Conversely, during the O-step, we subtract $\Phi_{prompt}$ to reduce the difference in the prompted area. This way, when we binary classify the contrastive potential (Equ 3), the point-prompted area will be classified as part of the object area.

### D.2   Adversarial Mask Updating

As discussed in Sec 4.1 and Fig 2, the I-step is capable of handling false positives (FP) and the O-step for false negatives (FN). If no mask updating constraints are applied, as the mask updating may not be accurate in the first several steps, new FP can be introduced in I-step. However, the new FP cannot

be addressed in the following O-step and has to wait for the next I-step to be handled. In contrast, if we constrain the I-step only shrinks the object mask, no FP will be introduced in the I-step and the remaining FN can be addressed in the following O-step. Similarly, we constrain O-step only to expand the object mask thus no FN will be introduced. We refer to the "shrink" in I-step and "expand" in O-step as adversarial mask updating. By employing the adversarial mask updating, we can address errors (FP/FN) that occurred during each step immediately without degradation.

### D.3 Expectation-Maximization

In PaintSeg, we introduce a latent variable $I_{paint}$ which is characterized by an off-the-shelf generative model $p(I_{paint}|I \circ M)$ conditioned on an image $I$ and a mask $M$. $\circ$ represents Hadamard product. In our method, we leverage the AMCP process to estimate and convert the latent variable $I_{paint}$ into mask prediction $M$ with alternating I-step and O-step. Mathematically, both I-step and O-step can be formulated as an expectation-maximization-like process.

- **Expectation**: We introduce a latent variable $I_{paint}$ in the proposed PaintSeg which is modeled by an off-the-shelf generative painting model $p(I_{paint}|I \circ M)$. We assume the generative model will pick the most likely outcome $I_{paint}$ given $I$ and $M$ for every step.

- **Maximization**: After obtaining the latent variable $I_{paint}$, we define a contrastive potential $\Phi$ and utilize clustering to binarize the mask. Mathematically, the contrasting and clustering processes maximize a posteriori probability $p(M|I_{paint}, I) = e^{-\frac{1}{\|M\|_0}\Phi(I_{paint},I,M)}$.

Although we term I-step and O-step separately, they can be formulated as the same EM process. PaintSeg advances the predicted mask to the ground truth by iteratively conducting the EM process in each step.

### D.4 Contrastive Potential

There are three terms in the contrastive potential $\Phi_{paint}$, $\Phi_{color}$ and $\Phi_{prompt}$:

- $\Phi_{paint}$: As analyzed in Sec 4.1, we can segment the object by contrasting the original and in/outpainted images. In this way, we formulate $\Phi_{paint}$ to describe the difference between the original and in/outpainted images and maximize it to locate the object.

- $\Phi_{color}$: As an image projector, i.e., DINO, is leveraged, the object segmentation from solely $\Phi_{paint}$ is achieved at a patch level. Therefore, we further introduce a color potential $\Phi_{color}$ to incorporate pixel-level color information to better segment the object boundary. $\Phi_{color}$ describes the probability of a pixel belonging to the foreground or background region based on the color similarity. By combining $\Phi_{paint}$ and $\Phi_{color}$, we can achieve high-quality object segmentation with AMCP.

- $\Phi_{prompt}$: In addition, the $\Phi_{prompt}$ is introduced to provide additional location information to guide the PaintSeg to locate the target object. We will add the discussion in the revision.

## E  Difference with Previous Segmentation Approaches

In this section, we discuss the major differences between the proposed PaintSeg and previous object segmentation methods as follows.

**Discriminative *v.s.* Generative + Discriminative .**  Conventional object segmentation is a discriminative task that leverages a neural network $\theta$ to model the conditional probability of the object mask $M$ given the image $I$ as condition $p_\theta(M|I)$. In PaintSeg, we have mask, paint, and contrast operations in each step. Specifically, in paint operation, we enroll a generative model to estimate painted image $I_{paint}$ with mask $M$ and image $I$ as conditions. After that, the mask can be obtained by comparing the generated image with the original one with a contrastive potential $\Phi$. As discussed in Section D.3, the paint operation is a generative process to estimate latent variable $p(I_{paint}|I \circ M)$ and the contrast operation is a discriminative process to obtain a mask prediction based on $p(M|I_{paint}, I)$. PaintSeg achieves training-free by constructing a bridge to generative painting models which permits object shape consistency and background content consistency.

**Pixel *v.s.* Pixel difference.** Conventional object segmentation leverages a network to project an image to the feature space and then binarize (cluster) each pixel into foreground or background classes. Differently, instead of directly clustering over the input image, PaintSeg utilizes the difference between the painted and original image, as a proxy, to leverage the object shape prior and background consistency. The contrastive scheme is rooted in the decomposable nature of images and paves a way to incorporate generated images to segment objects.

**Training *v.s.* Training-free.** Conventional object segmentation approaches train the neural network to segment objects requiring time-consuming and expensive data labeling. Some unsupervised segmentation methods [4, 10, 5, 61] find a segment from a generative model while they typically require training a network on top of the generative model. Instead, our method is a training-free unsupervised method that learns to segment objects from a generative painting model. We consider the PaintSeg provides a way to bridge the generative model and segmentation which may inspire future research.

# F   Failure Case Analysis

We analyze the failure case here. As shown in Fig. C, we visualize a failure case when using a point as the prompt. We notice the adjacent car is segmented as a false positive, which is due to the semantic and visual similarity between the target and false positive cars. Despite our method is capable of handling multiple objects with point prompt (right of Fig. C), crowded scenarios can make it difficult to segment the accurate object boundary. However, the issue can be overcome through box prompt.

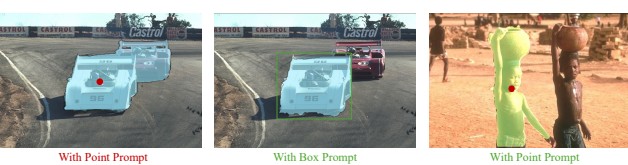

Figure C: Illustration of failure case.

# G   More Visualization

In this section, we demonstrate more visualization of PaintSeg. We show more qualitative results with box prompt in Fig. D, with point prompt in Fig. E and with coarse mask prompt in Figs. F and G.

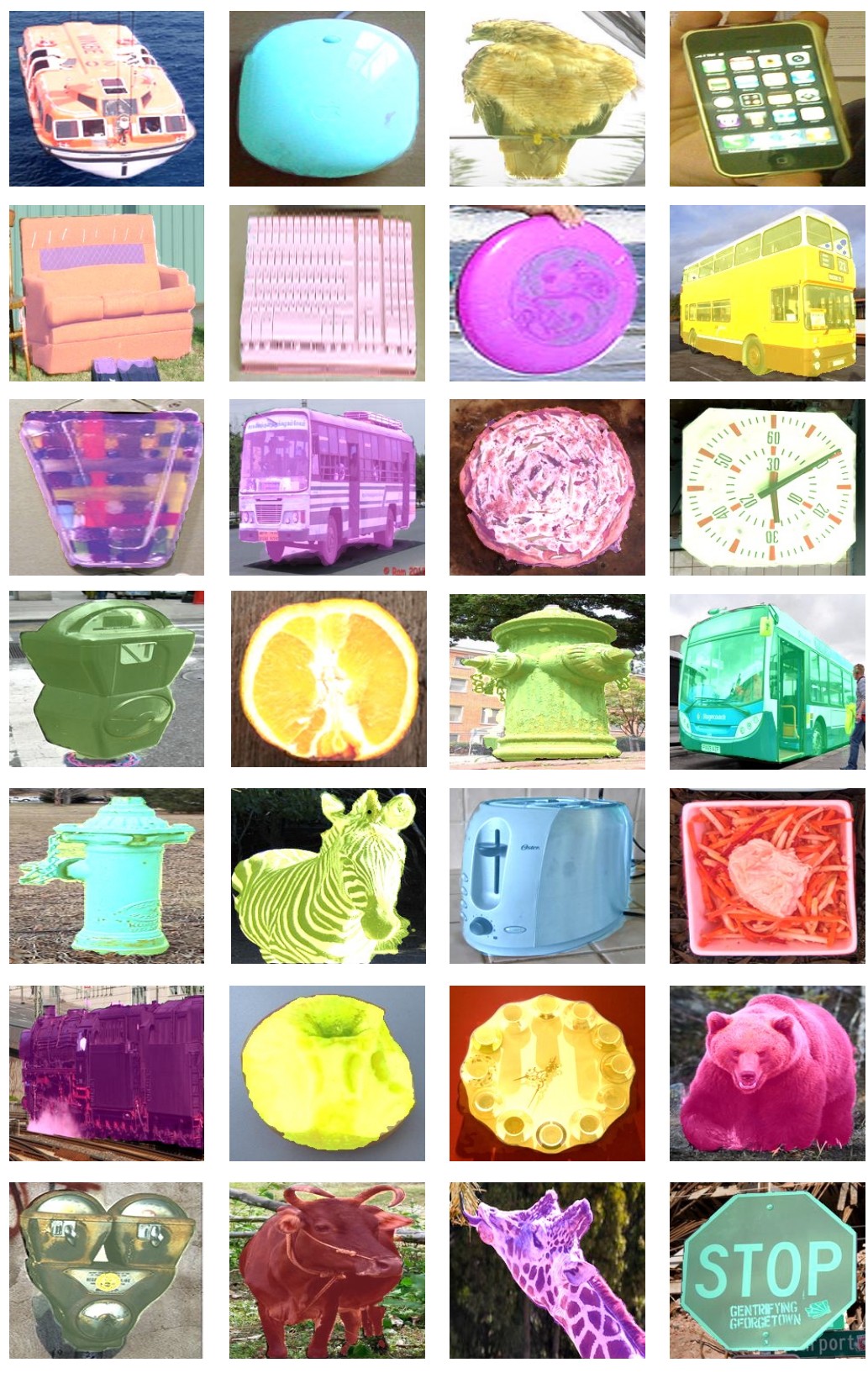

Figure D: **More visualization of PaintSeg with box prompt on COCO MVal.**

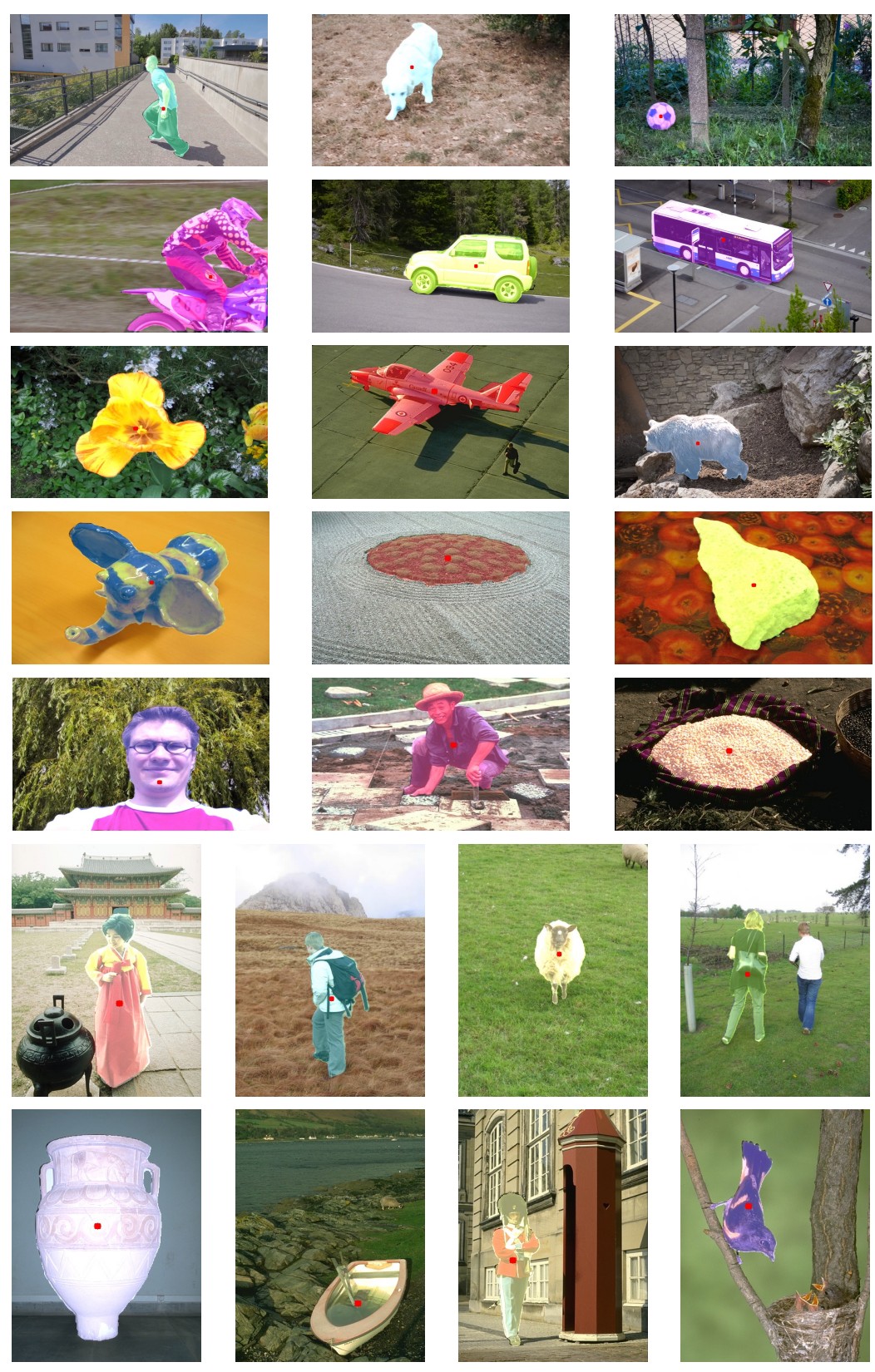

Figure E: **More visualization of PaintSeg with point prompt. The point prompt is illustrated by the red point on the image on DAVIS and Berkeley and GrabCut.**

| Prompt | Mask | Prompt | Mask |

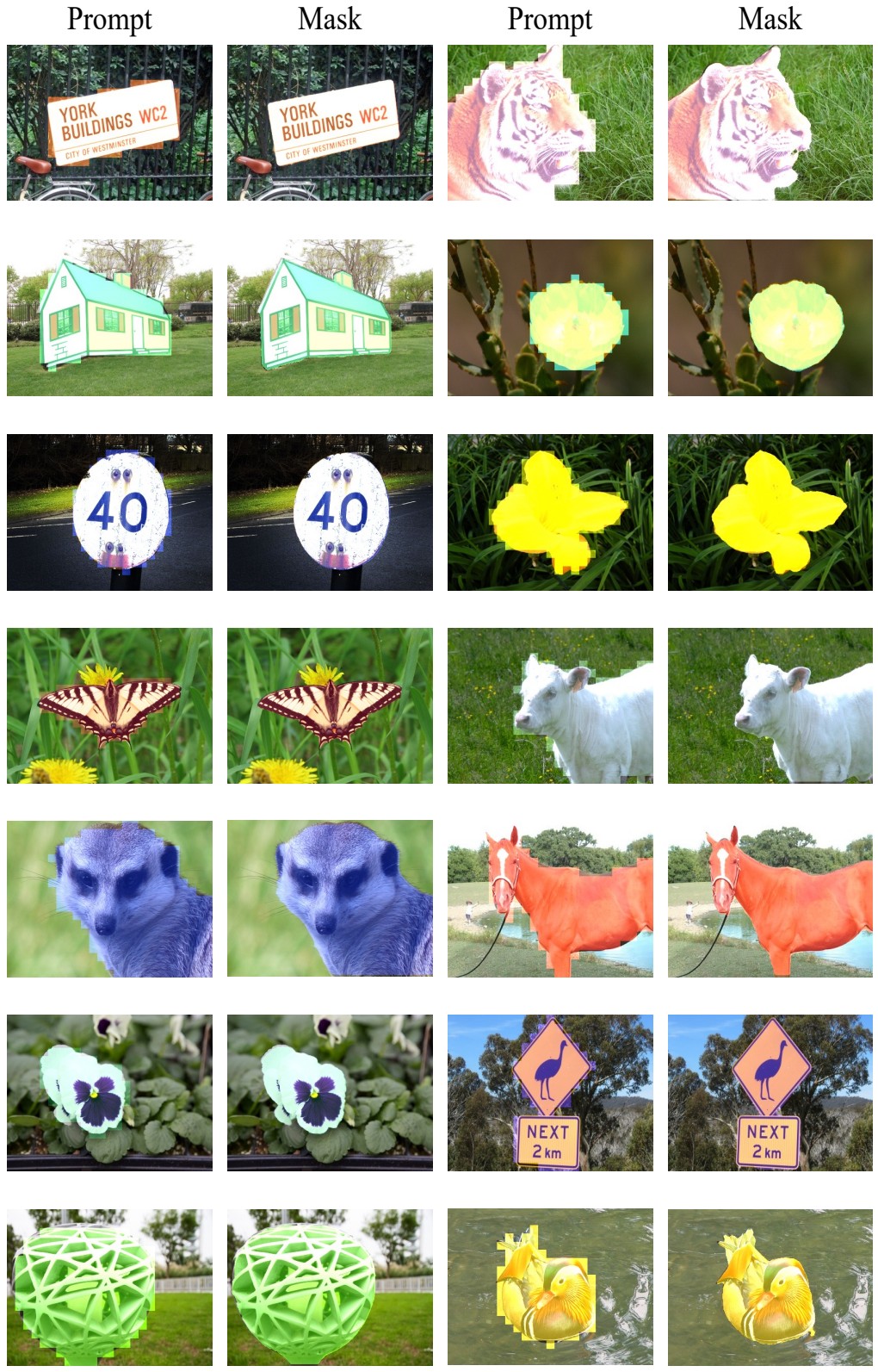

Figure F: **More visualization of PaintSeg with coarse mask prompt on ECSSD.**

| Prompt | Mask | Prompt | Mask |
|--------|------|--------|------|

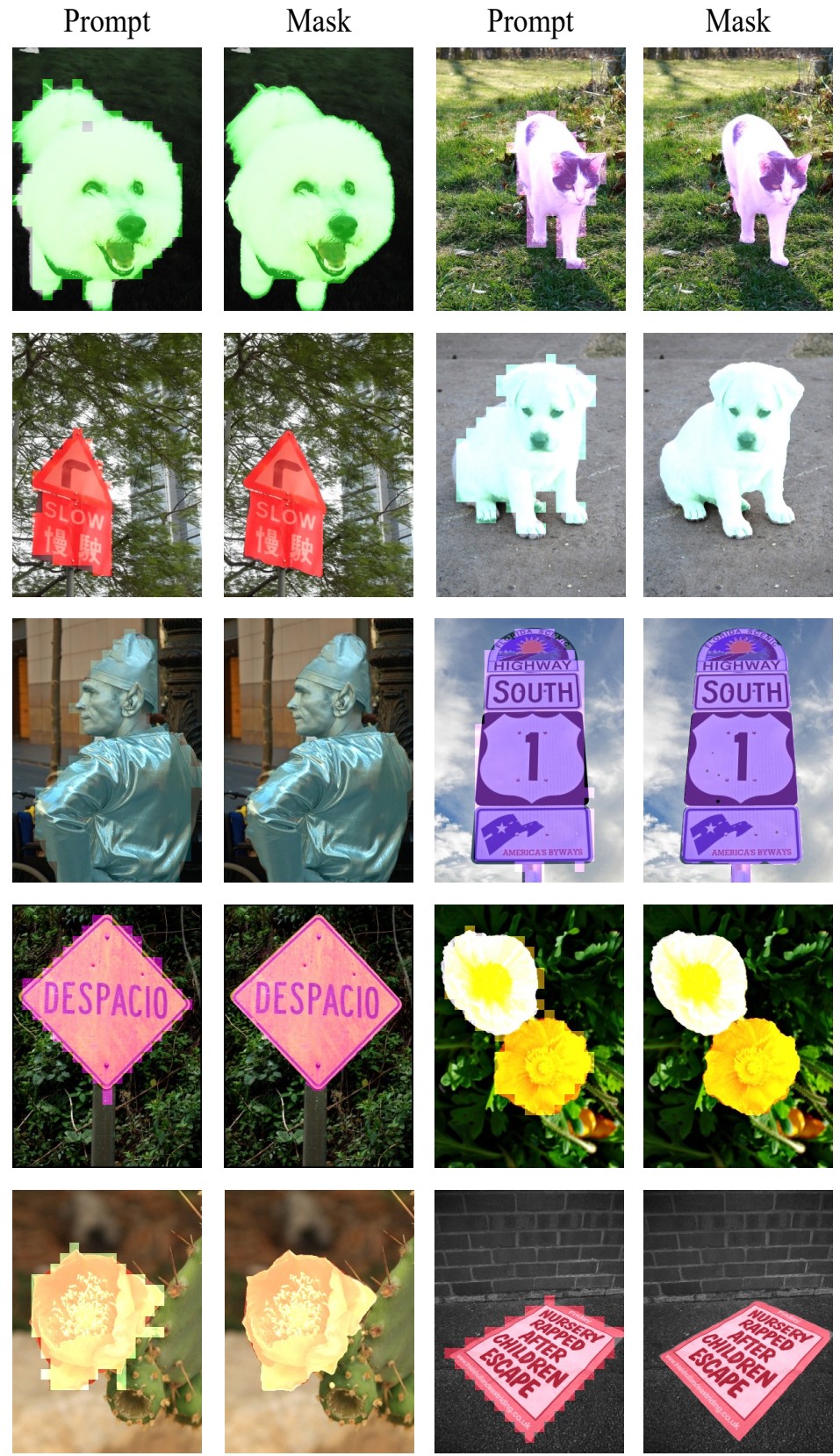

Figure G: **More visualization of PaintSeg with coarse mask prompt on ECSSD.**

