# OpenReview forum: "PaintSeg: Painting Pixels for Training-free Segmentation"
_NeurIPS.cc/2023/Conference — NeurIPS 2023 poster_

### Official Review · Reviewer_1n6h · 2023-06-16

**Soundness:** 3 good
**Presentation:** 2 fair
**Contribution:** 3 good
**Rating:** 6
**Confidence:** 4

**Summary:**

This paper proposed to interactively use inpainting (I-step) and outpainting (O-step) to achieve foreground masks without training.
Formally, the I-step refines the false positive masks while the O-step tackles false negative ones. The authors proposed Contrastive Potential to evaluate the differences between the origin and inpainted images, and gradually refine the segmentation.
Various prompts are supported to start the I-step and O-step (point, box, scribble).
Experiment results show the effectiveness of the proposed method.

**Strengths:**

1. Leveraging inpainting and outpainting to address the segmentation is novel and interesting.
2. Quantitative results show the effectiveness of the proposed method. It enjoys good improvements for untrained algorithms.
3. The mathematical formulation of EM is interesting.

**Weaknesses:**

1. I think this method is not efficient because it used a diffusion-based method to inpaint/outpaint iteratively. But the authors did not provide a comparison of efficiency.
2. Though the authors claim that their method could work with prompts: point, box, and scribble, they did not provide sufficient illustration about the usage of point prompts in Fig.2 and Sec4. In my opinion, both inpainting and outpainting fail to achieve proper results based on only one point prompt.
3. There are not enough ablations to confirm the effectiveness of Contrastive Potential (paint, color, prompt). It is unclear how to formulate Eq. (2) in the absence of point prompts.
4. Eq.(3) is a bit difficult to understand. It seems like $\mu$ is supposed to represent the center position of a cluster, but I'm not sure what the notation "$argmax(\mu_k)$" means. Personally, I believe it would be more helpful if the authors presented the average value of all the samples in a cluster as the value for $\mu," instead of just the cluster center.
5. A more equitable experiment would be to classify the competitors in Tab.2 into those with prompts and those without. It would be unjust to compare the proposed method with competitors lacking prompts.

**Questions:**

This paper enjoys good novelty. However, some presentations ($\mu_k$, point prompt) and experiments (efficiency, Contrastive Potential's ablation) are clear enough. I would increase my score if the authors could reasonably clarify them.

**Limitations:**

Yes the authors have discussed about them

---

> ### Author Rebuttal · Authors · 2023-08-07
>
> We thank the reviewer for the time and effort to review our paper. Our answers to the questions are as follows.
>
> ---
>
> **1. The method is not efficient. Lack of comparison of efficiency.**
>
> We acknowledge that PaintSeg is not efficient. We report the computing time for each component of AMCP tested with NVIDIA Tesla V100 32G GPU in Tab A. We notice that the majority of time is consumed by the diffusion model.
>
> As shown in Tab B, we ablate on the impact of different generative models on PaintSeg. For the generative model with similar performance (SD1.5, SD2.1, Kandinsky 2.1), we notice that the performance of PaintSeg remains robust. This suggests that PaintSeg holds promise for facile integration with various generative models in a plug-and-play manner. Considering the swift advancements in diffusion sampling [R1,R2], we anticipate that the speed of PaintSeg can be further enhanced in the future by leveraging more advanced diffusion models.
>
> | In/Outpainting (diffusion) | Calculating $\Phi$ (CRF) | K-means | Total |
> |:----:| :----: | :----: | :----: |
> | 13.91s | 0.46s | 0.02s | 14.39s |
>
> Table A: Latency of components for each iteration of AMCP.
>
> |Generative model| Mask | Box |
> |:----| :----: | :----: |
> | Latent diffusion | 76.8 | 68.4 |
> | Kandinsky 2.1| 79.4 | 71.5 |
> | Stable diffusion 2.1 | 80.1 | 72.2 |
> | Stable diffusion 1.5 | 80.6 | 71.0 |
>
> Table B: Ablation of generative models on ECSSD dataset.
>
> [R1] Consistency Models, ICML 2023
>
> [R2] One-Step Diffusion Distillation via Deep Equilibrium Models, ICML 2023 Workshop
>
> ---
>
> **2. Not sufficient illustration about the usage of point prompts. In my opinion, both inpainting and outpainting fail to achieve proper results based on only one point.**
>
> Sorry for the confusion. We would like to clarify that, the entire image area is set to be the init mask for point and scribble prompts (Line 233-235), and the prompt term $\Phi_{prompt}$ in contrastive potential $\Phi$ serves as the guidance to locate the object. In this way, we can avoid outpainting with a small conditioned area. We visualize the process of point-prompted AMCP in Fig A (in the PDF file within the general response).
>
> We notice that Fig 2 might be misleading as we illustrate the rationale of contrastive painting with examples of box and scribble prompts. We will replace the "box" and "scribble" in Fig 2 with "mask with false positives" and "mask with false negatives" in the revision.
>
> ---
>
> **3. Ablation of contrastive potential. How to formulate it with point prompts?**
>
> We conduct an ablation study on contrastive potential as shown in Tab C. Compared to solely leverage $\Phi_{paint}$, combining $\Phi_{paint}$ and $\Phi_{color}$ yields better performance by refining the patch-level mask to pixel-level. For $\Phi_{prompt}$, it provides additional location information thus benefits the performance. By initializing the entire image as the input mask for point and scribble prompts, the contrastive potential for them can be calculated in the same way as box and mask prompts.
>
> | $\Phi_{paint}$ | $\Phi_{color}$ | $\Phi_{prompt}$ | Mask | Box | Point |
> | ----| :----: | :----: | :----: | :----: | :----: |
> | &#10004; |                 |                 | 71.4 | 67.8 | - |
> | &#10004; | &#10004; |                 | 80.6 | 70.2 | - |
> | &#10004; |                 | &#10004; | -      | 68.1 | 56.9 |
> | &#10004; | &#10004; | &#10004; | -      | 71.0 | 60.8 |
>
> Table C: Ablation of the components in the contrastive potential. For the mask prompt, $\Phi_{prompt}$ is not used (Line 156-157) and, for the point prompt, $\Phi_{prompt}$ is necessary to locate the object.
>
> ---
>
> **4. The meaning of $\mu$. Better to present the average value of all the samples in a cluster as the value for $\mu$ instead of just the cluster center.**
>
> Thanks for your suggestion! Yes, it would be more clear to describe $\mu$ as the average value of all samples in a cluster. We will revise it in the revision.
>
> ---
>
> **5. Better to classify competitors into those with/without prompts. Unjust to compare the proposed method with competitors lacking prompts in Tab 2.**
>
> Thank you for your suggestion. We consider that all methods in Tab 2 are equipped with the same point prompt (centroid of the GT mask, following convention) including training-free methods such as GrowCut [54]. Therefore, we believe the comparison in Tab 2 is conducted fairly.

---

> > ### Comment · Reviewer_1n6h · 2023-08-12
> > **Thanks for the response.**
> >
> > Thanks for the response from the authors. After reading the rebuttal, most of my concerns are addressed.
> > But I am still confused about the point prompt. While the whole image is masked, how to use a point to locate the object?
> > Unfortunately, I think that FigA(point) is similar to Fig4(box). Without any prompt annotation in the figure, it is difficult to understand the role of "point prompting", and what is the difference between point and box.

---

> > > ### Author Response · Authors · 2023-08-12
> > > **Response to Reviewer 1n6h**
> > >
> > > Thank you for your question. We would like to provide a more detailed clarification of the role of point prompting.
> > >
> > > As depicted in Fig 2, the object mask can be derived by contrasting in/outpainted images with the original ones. In the I-step, there is a substantial difference (Fig 2 (a)) in the object area. In contrast, in the O-step, the object area has a small difference (Fig 2 (b)).
> > >
> > > To incorporate the point information, we introduce a prompt term $\Phi_{prompt}$ in the contrastive potential to enhance the difference based on the point location. Specifically, $\Phi_{prompt}[i,j] = \mathcal{G}(x,y)[i,j]$ (Equ 2) represents a gaussian offset. Here, $(x,y)$ refers to the point prompt, serving as the mean, while $\frac{1}{10}$ of the bounding box's height and width determine the variance (Line 159).  During the I-step, we add $\Phi_{prompt}$ to the contrastive potential to amplify the difference in the prompted area. Conversely, during the O-step, we subtract $\Phi_{prompt}$ to reduce the difference in the prompted area. This way, when we binary classify the contrastive potential (Equ 3), the point-prompted area will be classified as part of the object area.
> > >
> > > Both box and point prompts follow the same process, differing in their initial masks and Gaussian offsets. To summarize the distinctions:
> > >
> > > | Prompt | Initial Mask | Mean of Guassian offset |
> > > | ----| :---- | :---- |
> > > | Box | Box area | Center of the box (Line 160) |
> > > | Point | Entire image | Coordinate of the point (Line 157) |
> > >
> > > Table D: Difference between point and box prompts.
> > >
> > > In our revisions, we will generate separate visualizations of the contrastive potential to better illustrate the function of each term. We hope the responses could clarify your confusion, and we are more than happy to provide further explanations. Thank you.

---

> > > > ### Comment · Reviewer_1n6h · 2023-08-12
> > > > **Thanks for the response.**
> > > >
> > > > Thanks. Since Fig4 said that it only visualized the box prompted area, it would be better to provide the whole image on the left with the framed target.
> > > > I would raise my score to vote for acceptance.

---

> > > > > ### Author Response · Authors · 2023-08-12
> > > > > **Thank you for your valuable feedback**
> > > > >
> > > > > Thank you for your valuable feedback! We will incorporate the suggested changes into our revision.

---

### Official Review · Reviewer_CwUQ · 2023-07-01

**Soundness:** 3 good
**Presentation:** 3 good
**Contribution:** 2 fair
**Rating:** 6
**Confidence:** 4

**Summary:**

The author proposes a training-free segmentation technique AMCP, which leverages off-the- shell diffusion model to refine the corase segmentation prompt iteratively. AMCP contains three main components: Inpainting step, outpainting step and adversarial updating. The paper gives detail ablation of AMCP and demonstrate its effectiveness on sevaral benchmarks.

**Strengths:**

1. This paper achieves a successful integration of an existing denoising diffusion model, exploiting its potential in segmentation tasks by inpainting and outpainting techniques, resulting in promising performance.
2. This paper conducts comprehensive comparisons with other segmentation approaches and provides a detailed ablation analysis of the techniques employed in AMCP

**Weaknesses:**

1. The main contribution to the segmentation ability stems from the diffusion model. However, the low speed of the sampling process in the diffusion model poses a significant obstacle for real-time applications. As AMCP lacks classification and object discovery capabilities, it is better suited as a labeling tool or an interactive segmenter, the latter of which demands even higher real-time performance. However, achieving the performance demonstrated in the paper requires AMCP to undergo several diffusion steps (5), further diminishing its effectiveness.

2. I notice that AMCP is sensitive to the cluster center number, which leads to about 20 mIoU variations. In the paper, AMCP is fixed to use the pre-training parameters of one specific diffusion model. When employing different diffusion models, the impact of each inpainting varies. Whether the hyperparameters of AMCP reasoning also need to be adjusted accordingly.

**Questions:**

1. The contrastive potential consists of three items. How does each item affect the segmentation result？

**Limitations:**

Authors address the limitations of their work and potential negative societal impact.

---

> ### Author Rebuttal · Authors · 2023-08-07
>
> We thank the reviewer for the time and effort to review our paper. Our answers to the questions are as follows.
>
> ---
>
> **1. The segmentation ability stems from the diffusion model while the sampling speed of the diffusion model obstacles for real-time applications.**
>
> We acknowledge that PaintSeg is not efficient. While PaintSeg actually relies on high-quality in/outpainted images, without necessitating a specific generative model. Considering the swift advancements in diffusion sampling [R1, R2], we anticipate that the speed of AMCP can be further enhanced in the future by leveraging more advanced diffusion models.
>
> In addition to the value of applications, PaintSeg also reveals that satisfactory segmentation can be achieved in a fully unsupervised manner by contrasting original and in/outpainted images which can be of interest to the community.
>
> [R1] Consistency Models, ICML 2023
>
> [R2] One-Step Diffusion Distillation via Deep Equilibrium Models, ICML 2023 Workshop
>
> ---
>
> **2. Performance is sensitive to cluster center numbers. When employing different diffusion models, whether the hyperparameters of AMCP reasoning also need to be adjusted accordingly?**
>
> Since only 2 categories - foreground and background are required to discriminate, a cluster number of 2 is the most reasonable choice. We ablate on the cluster number to verify if an ambiguous region between foreground and background exists when calculating contrastive potential $\Phi$. As AMCP with a cluster number larger than 2 will discard an ambiguous region in each step, the performance will largely drop if the ambiguous region does not exist (we verify that a cluster number of 2 leads to the best result).
>
> We consider that the core of AMCP is based on contrasting in/outpainted and original images and the hyperparameters of AMCP do not have a model-specific design. In this way, if the generated image quality of the diffusion model does not have a big gap, no further hyperparameter adjustment is required. As shown in Tab A, we ablate on different generative models. For the generative model with similar performance (SD1.5, SD2.1, Kandinsky 2.1), we notice that AMCP is robust with fixed hyperparameters. However, for the diffusion model with obviously inferior performance (latent diffusion), AMCP performance will degrade accordingly.
>
> |Generative model| Mask | Box |
> |:----| :----: | :----: |
> | Latent diffusion | 76.8 | 68.4 |
> | Kandinsky 2.1| 79.4 | 71.5 |
> | Stable diffusion 2.1 | 80.1 | 72.2 |
> | Stable diffusion 1.5 | 80.6 | 71.0 |
>
> Table A: Ablation of generative models on ECSSD dataset.
>
> ---
>
> **3. How does each item in contrastive potential affect the segmentation result?**
>
> There are three terms in the contrastive potential $\Phi_{paint}$, $\Phi_{color}$ and $\Phi_{prompt}$.
>
> - $\Phi_{paint}$: As analyzed in Sec 4.1, we can segment the object by contrasting the original and in/outpainted images. In this way, we formulate $\Phi_{paint}$ to describe the difference between the original and in/outpainted images and maximize it to locate the object.
>
> - $\Phi_{color}$: As an image projector, i.e., DINO, is leveraged, the object segmentation from $\Phi_{paint}$ is achieved at a patch level. Therefore, we further introduce a color potential $\Phi_{color}$ to incorporate pixel-level color information to better segment the object boundary. $\Phi_{color}$ describes the probability of a pixel belonging to the foreground or background region based on the color similarity. By combining $\Phi_{paint}$ and $\Phi_{color}$, we can achieve high-quality object segmentation with AMCP.
>
> - $\Phi_{prompt}$: In addition, the $\Phi_{prompt}$ is introduced to provide additional location information to guide the PaintSeg to the target object.
>
> We conduct an ablation study on components in contrastive potential as shown in Tab B. Compared to solely leverage $\Phi_{paint}$, combining $\Phi_{paint}$ and $\Phi_{color}$ yields better performance by refining the patch-level mask to pixel-level. For $\Phi_{prompt}$, it provides additional location information thus benefits the performance.
>
> | $\Phi_{paint}$ | $\Phi_{color}$ | $\Phi_{prompt}$ | Mask | Box | Point |
> | ----| :----: | :----: | :----: | :----: | :----: |
> | &#10004; |                 |                 | 71.4 | 67.8 | - |
> | &#10004; | &#10004; |                 | 80.6 | 70.2 | - |
> | &#10004; |                 | &#10004; | -      | 68.1 | 56.9 |
> | &#10004; | &#10004; | &#10004; | -      | 71.0 | 60.8 |
>
> Table B: Ablation of the components in the contrastive potential. For the mask prompt, $\Phi_{prompt}$ is not used (Line 156-157) and, for the point prompt, $\Phi_{prompt}$ is necessary to locate the object (Line 234-235).

---

> > ### Comment · Reviewer_CwUQ · 2023-08-20
> >
> > Thanks for the response. The authors have addressed most of my concerns, and I would like to increase my rating. I suggest that the discussion of model efficiency could be incorporated into the revised version.

---

> > > ### Author Response · Authors · 2023-08-20
> > >
> > > Thank you for your valuable comments. We will incorporate the suggested changes into our revision.

---

### Official Review · Reviewer_9reF · 2023-07-03

**Soundness:** 3 good
**Presentation:** 2 fair
**Contribution:** 4 excellent
**Rating:** 6
**Confidence:** 4

**Summary:**

The following work proposes an iterative approach to converting any off-the-shelf inpainting/outpainting capable models to perform segmentation. The proposed method alternates between two approaches to separating the segmentable foreground object from the background. In the I(npainting) step, they mask out the foreground object and inpaint the gap, then locally contrast the in-painted image against the original. The inpainting will fill the hole with an extension of the background instead of the masked out foreground object, thus the disparity between the two images gives rise to a segmentation mask for said object. The O(utpainting) step is similar but expands outwards from a sample of the foreground object. The disparity between the outpainted image and the original image will be large on the non-foreground pixels, resulting in an approximate inverted result from the I-step.  Experiments suggest significant improvements over recent methods with limited to no additional training.

**Strengths:**

- Approach appears principled and builds upon older methods involving conditional random fields
- Strong results with minimal (or none at all?) additional supervised training necessary; potential for strong generalization

**Weaknesses:**

To me, the primary concerns of this work lie in the clarity of the writing:
- While I understand why the authors chose to refer to the approach as adversarial masking, I don't think this is correct. The inpainting and outpainting steps come from opposite direction but have a more complimentary relationship than an adversarial one. The formulation in (6) similarly does not exhibit signs of any adversarial interaction between opposing objectives, but would be more aptly described an alternating minimization scheme of two separate constraint terms.
- Overall, I felt that the writing could have been more clear. In section 2 on contrastive potentials, the authors assign the sum of potential functions to the term contrastive potentials. Critically missing is how the potential values relate to the segmentation task itself, presumably through an argmax obejctive. In addition, while the terms inpainting and outpainting are generally accepted terminology, I don't believe it is common to refer to the resulting image as the "painted image", which definitely led to some reading confusion on my end.
- It is also unclear to me where the cluster centers \mu_k come from in the k-means mask update step in section 4.3
- Some additional missing ablations include ablations over the lambda weights for individual potential functions, as well as choice of generative model.

**Questions:**

Address weakness section

**Limitations:**

Adequately addressed

---

> ### Author Rebuttal · Authors · 2023-08-07
>
> We thank the reviewer for the time and effort to review our paper. Our answers to the questions are as follows.
>
> ---
>
> **1. Does PaintSeg require supervision?**
>
> In the entire pipeline, PaintSeg does not require any supervision. We find it intriguing to demonstrate that satisfactory segmentation can be achieved in a fully unsupervised manner by contrasting original and inpainted/outpainted images
>
> ---
>
> **2. The term "adversarial masking" is not appropriate as the relation between inpainting and outpainting is more complementary than adversarial.**
>
> We would like to clarify that the "adversarial mask updating" does not refer to the alternating I- and O-steps but the mask updating constraints (Equ 4: only shrink mask in I-step and expand mask in O-step). As discussed in Sec 4.1 and Fig 2, the I-step is capable of handling false positives (FP) and the O-step for false negatives (FN).
>
> - *Without mask updating constraints*: Since mask updating may lack accuracy in the initial steps, new FPs can emerge after the I-step. However, these new FPs cannot be rectified in the subsequent O-step and must await the next I-step for resolution. Similarly, new FNs can emerge during the O-step.
>
> - *With mask updating constraints*: By constraining the I-step solely to "shrink" the object mask, new FPs will not emerge in the I-step. The remaining FNs can then be addressed in the following O-step. Similarly, by limiting the O-step to "expand" the object mask, new FNs will not be introduced.
>
> We refer to the "shrink" in I-step and "expand" in O-step as adversarial mask updating. By employing the adversarial mask updating, we can promptly address errors (FP/FN) that arise during each step without degradation. We will further clarify the term "adversarial" in the revision.
>
> ---
>
> **3. How do contrastive potential values relate to the segmentation itself? The term "painted image" is confusing.**
>
> There are three terms in the contrastive potential $\Phi_{paint}$, $\Phi_{color}$ and $\Phi_{prompt}$.
>
> - $\Phi_{paint}$: As analyzed in Sec 4.1, we can segment the object by contrasting the original and in/outpainted images. In this way, we formulate $\Phi_{paint}$ to describe the difference between the original and in/outpainted images and maximize it to locate the object.
>
> - $\Phi_{color}$: As an image projector, i.e., DINO, is leveraged, the object segmentation from $\Phi_{paint}$ is achieved at a patch level. Therefore, we further introduce a color potential $\Phi_{color}$ to incorporate pixel-level color information to better segment the object boundary. $\Phi_{color}$ describes the probability of a pixel belonging to the foreground or background region based on the color similarity. By combining $\Phi_{paint}$ and $\Phi_{color}$, we can achieve high-quality object segmentation with AMCP.
>
> - $\Phi_{prompt}$: In addition, the $\Phi_{prompt}$ is introduced to provide additional location information to guide the PaintSeg to the target object.
>
> Thank you for pointing out the confusion about the "painted image". We will revise the term as "in/outpainted image" in the revision to enhance the readability.
>
> ---
>
> **4. Where does cluster center $\mu_k$ come from k-means?**
>
> Since we conduct k-means on the contrastive potential values $\Phi[h,w]\in\mathbb{R}^{1}$, the cluster center is the average value of all the samples in a cluster. For the 2-clusters case, the $\mu_k$ refers to the average value of $\Phi$ in the foreground or background regions. We will revise $\mu_k$ as the average value of all the samples in the $k$- cluster in our revision.
>
> ---
>
> **5. Ablation on lambda weights for individual potential functions and the choice of the generative model.**
>
> We conduct the ablation study to investigate the influence of potential function as shown in Tab A. Additionally, we ablate the weights to combine $\Phi_{paint}$ and $\Phi_{color}$ as shown in Tab B. We notice each term in the contrastive potential benefits the performance.
>
> As shown in Tab C, we ablate on different generative models. For the generative model with similar performance (SD1.5, SD2.1, Kandinsky 2.1), we notice that the performance of PaintSeg is robust. However, if the diffusion model has obviously inferior performance (latent diffusion), AMCP performance will degrade accordingly.
>
> | $\Phi_{paint}$ | $\Phi_{color}$ | $\Phi_{prompt}$ | Mask | Box | Point |
> | ----| :----: | :----: | :----: | :----: | :----: |
> | &#10004; |                 |                 | 71.4 | 67.8 | - |
> | &#10004; | &#10004; |                 | 80.6 | 70.2 | - |
> | &#10004; |                 | &#10004; | -      | 68.1 | 56.9 |
> | &#10004; | &#10004; | &#10004; | -      | 71.0 | 60.8 |
>
> Table A: Ablation of the components in the contrastive potential. For the mask prompt, $\Phi_{prompt}$ is not used (Line 156-157) and, for the point prompt, $\Phi_{prompt}$ is necessary to locate the object.
>
> | $\lambda_{paint}$ | $\lambda_{color}$ | Mask |
> | ----| :----: | :----: |
> | 0.80 | 0.10 | 79.7 |
> | 0.80 | 0.15 | 80.1 |
> | 0.80 | 0.20 | 80.6 |
> | 0.80 | 0.25 | 80.4 |
>
> Table B: Ablation of the weights of contrastive potential on ECSSD.
>
> |Generative model| Mask | Box |
> |:----| :----: | :----: |
> | Latent diffusion | 76.8 | 68.4 |
> | Kandinsky 2.1| 79.4 | 71.5 |
> | Stable diffusion 2.1 | 80.1 | 72.2 |
> | Stable diffusion 1.5 | 80.6 | 71.0 |
>
> Table C: Ablation of generative models on ECSSD dataset.

---

> > ### Comment · Reviewer_9reF · 2023-08-19
> > **Concerns appropriately addressed**
> >
> > I would like to thank the authors for taking the time to address each of my concerns regarding the clarity of the paper. I am increasing my rating to a weak accept and urge the authors to incorporate the suggested changes into their writing to improve the readability of their work.

---

> > > ### Author Response · Authors · 2023-08-20
> > > **Thank you for your valuable feedback**
> > >
> > > We greatly appreciate your valuable feedback! We will incorporate the suggested changes into our revision.

---

### Official Review · Reviewer_fhrH · 2023-07-05

**Soundness:** 3 good
**Presentation:** 3 good
**Contribution:** 3 good
**Rating:** 7
**Confidence:** 4

**Summary:**

This paper presents a new unsupervised method for object segmentation, named PaintSeg. Given the initial mask, the proposed PaintSeg adopts adversarial masked contrastive painting (AMCP) to generate the foreground and background masks through the contrastive relations between the original image and the painted image. The AMCP process involves two steps: I-step for the foreground mask and O-step for the background mask. The proposed PaintSeg based on the off-the-shelf diffusion models does not rely on training with labeled samples but generate segmentation results through iterative optimization with the I-step and the O-step. Experiments show the good performance achieved by the proposed PaintSeg.

**Strengths:**

1. The idea of using inpainting and outpainting for foreground and background masks is interesting and novel to me.
2. This paper addresses unsupervised image segmentation in a training-free manner with pre-trained generative models.
3. This paper presents the optimization steps with contrastive potential and adversarial steps.
4. The proposed approach PaintSeg is effective and achieves good results on several benchmarks.


**Weaknesses:**

1. Firstly, it's a nice work and there is no significant weakness.
2. Based on LDM, the proposed PaintSeg takes much time to iteratively generate masks.


**Questions:**

1. Considering that the proposed PaintSeg adopts diffusion models for inpainting, I'm concerned about whether PaintSeg is sensitive to the generative models and whether different generative models affect the final quality or the convergence speed.
2. How about the latency of generating the segmentation mask.
3. I suggest the authors add the evaluation results of SAM with the same prompts, which can show the upper bound or the superiority of the proposed approach.

**Limitations:**

The authors have clarified the limitations.

---

> ### Author Rebuttal · Authors · 2023-08-07
>
> We thank the reviewer for the time and effort to review our paper. Our answers to the questions are as follows.
>
> ---
>
> **1. Whether PaintSeg is sensitive to the generative models?**
>
> We consider PaintSeg is not sensitive to generative models. PaintSeg segments objects by contrasting original and in/outpainted images. Even if different generative models may tend to generate objects in different styles, as long as the generated object follows the realistic distribution, PaintSeg can decode it with the EM-like process. As shown in Tab A, we conduct an ablation study to replace different generative models in AMCP. We notice that, for the generative model with similar performance (Stable diffusion1.5, SD2.1, Kandinsky 2.1), AMCP is robust with fixed hyperparameters. However, for the diffusion model with obviously inferior performance (latent diffusion), AMCP performance will degrade accordingly.
>
> |Generative model| Box |Mask |
> |:----| :----: | :----: |
> | Latent diffusion |  68.4 | 76.8 |
> | Kandinsky 2.1| 71.5 | 79.4 |
> | Stable diffusion 2.1 |  72.2 | 80.1 |
> | Stable diffusion 1.5 |  71.0 | 80.6 |
>
> Table A: Ablation of generative models on ECSSD dataset.
>
> ---
>
> **2. Latency of generating the segmentation mask?**
>
> We report the computing time of AMCP tested on NVIDIA Tesla V100 32G GPU as shown in Tab B. We notice the most time-consuming part is the multi-step diffusion inference. As AMCP is not sensitive to generative models, with the extensive research in diffusion sampling [R1,R2], we consider the speed of AMCP can be further improved with more advanced diffusion models in the future.
>
> | In/Outpainting (diffusion) | Calculating $\Phi$ (CRF) | K-means | Total |
> |:----:| :----: | :----: | :----: |
> | 13.91s | 0.46s | 0.02s | 14.39s |
>
> Table B: Latency of components for each iteration of AMCP.
>
> [R1] Consistency Models, ICML 2023
>
> [R2] One-Step Diffusion Distillation via Deep Equilibrium Models, ICML 2023 Workshop
>
> ---
>
> **3. Comparison with SAM.**
>
> We report the performance of SAM with the same prompts as used in PaintSeg in Tab C. We notice that SAM shows superior performance compared to PaintSeg which can be attributed to two main factors: (1) the large-scale supervised training with labeled data, and (2) the large number of parameters. Different from the supervised pipeline employed by SAM, PaintSeg demonstrates that satisfactory segmentation can be achieved in a fully unsupervised manner by contrasting original and in/outpainted images. We consider the findings of PaintSeg to remain valuable even if its performance is suppressed by SAM.
>
> | Method | GrabCut (point) | Berkeley (point) | DAVIS (point) |  PASCAL VOC (box) | MVal (box) |
> |:----| :----: | :----: | :----: | :----: | :----: |
> | SAM [29] | 86.7 | 83.3 | 71.1 | 66.4 | 83.0 |
> | PaintSeg| 84.4 | 70.0 | 69.4 | 59.7 | 69.6 |
>
> Table C: Comparison with SAM for point and box prompted segmentation.

---

> > ### Comment · Reviewer_fhrH · 2023-08-20
> >
> > The rebuttal from the authors has addressed my concerns. And I suggest the authors add those results to the revised version as they are more insightful. At last, I keep my rating.

---

> > > ### Author Response · Authors · 2023-08-20
> > >
> > > Thank you for your time and consideration. We will add the results and discussions in the revised version.

---

### Official Review · Reviewer_Xo6H · 2023-07-06

**Soundness:** 2 fair
**Presentation:** 2 fair
**Contribution:** 2 fair
**Rating:** 6
**Confidence:** 4

**Summary:**

This paper designs a novel unsupervised segmentation method (AMCP) without training via image painting. Its main process is to mask the image based on the specified prompt, recover the image using a generative model, and determine the foreground mask and background mask of the image by comparing the original and inpainted images. Specifically, based on the specified prompt, the image is first masked and then recover using a generative model. Finally, by comparing the original and inpainted images, the foreground mask and background mask of the image are determined. Specifically, it designed I-step and O-step respectively. In the I-step, the background of the image is restored from the deleted object region, and the foreground mask can be obtained by comparing it with the original image. Conversely, in the O-step, the background mask can be obtained by drawing the foreground in the image. Finally, a large number of experiments fully demonstrate the effectiveness of the method.

**Strengths:**

1. The idea of obtaining the segmentation result of an image by comparing the difference between the original and inpainted images is interesting.
2. The manuscript is well organized and has several experiments.

**Weaknesses:**

1. Mt+ and Mt− serve as the dilation and erosion masks of Mt, and are important conditions for updating the regions near the foreground-background boundary. The description in the implementation details cannot show the process and loses the discussion of their necessity.
2. Since AMCP first performs the I-step and the input init mask needs to include the false positives region, it is not possible to directly perform the I-step for point or scribble prompts. In this case, is it feasible to swap the order of the I-step and O-step?
3. DINO and Fully Connected CRF are used in Line 150 and 154 respectively, and they play a very important role in AMCP. What would be the impact on AMCP if they were replaced? DINO's features are highly robust, and replacing DINO with other models can be used to verify if it is DINO that has improved the upper limit of AMCP.
4. The core of this paper is to segment images based on a specific prompt, which is the same setting as SAM and SEEM. Therefore, fair comparisons should be made with them in comparative experiments. Currently, the comparison method introduced in the paper is not fair, as PaintSeg requires a specific prompt to be given, while most of the unsupervised methods being compared do not have this setting.
5. Although AMCP is a training-free method, it involves many hyperparameters and other introduced models. It is recommended to add pseudocode to more clearly demonstrate the detailed process of AMCP.

**Questions:**

shown in weaknesses

**Limitations:**

shown in weaknesses

---

> ### Author Rebuttal · Authors · 2023-08-07
>
> We thank the reviewer for the time and effort to review our paper. Our answers to the questions are as follows.
>
> ---
>
> **1. The role and necessity of updating near the foreground-background boundary (adversarial mask updating).**
>
> We would like to further clarify the necessity of adversarial mask updating. As discussed in Sec 4.1 and depicted in Fig 2, the I-step effectively addresses false positives (FP) and the O-step for false negatives (FN).
>
> - *Without mask updating constraints*: Since mask updating may lack accuracy in the initial steps, new FPs can emerge after the I-step. However, these new FPs cannot be rectified in the subsequent O-step and must await the next I-step for resolution. Similarly, new FNs can emerge during the O-step.
>
> - *With mask updating constraints*: By constraining the I-step solely to "shrink" the object mask, new FPs will not emerge in the I-step. The remaining FNs can then be addressed in the following O-step. Similarly, by limiting the O-step to "expand" the object mask, new FNs will not be introduced.
>
> We refer to the "shrink" in I-step and "expand" in O-step as adversarial mask updating. By employing the adversarial mask updating, we can promptly address errors (FP/FN) that arise during each step without degradation.
>
> The terms "dilation" and "erosion" in Section 5.2 (implementation details) pertain not to adversarial mask updating, but to the morphological operations employed for filtering sparse points after binarizing the contrastive potential into the object mask. We will revise the wording in the revised version."
>
> ---
>
> **2. Since AMCP first performs the I-step and the input init mask needs to include the false positives region, it is not possible to directly perform the I-step for point or scribble prompts. Is it feasible to swap the order of I-step and O-step for point and scribble prompts?**
>
> Upon initially performing the O-step using point and scribble prompts, we observed that the quality of the outpainted image lacks robustness due to the limited conditioned area within the point/scribble regions. To address this, we establish the entire image area as the initial mask for point and scribble prompts (Lines 233-235) and initiate AMCP from the I-step. Object localization is guided by the prompt term $\Phi_{prompt}$ in the contrastive potential $\Phi$. The process of point-prompted AMCP is visualized in Fig A (found in the PDF file within the general response).
>
> We recognize that Fig 2 might be misleading since it illustrates the rationale behind contrastive painting using examples of box and scribble prompts. In the revision, we will replace the terms "box" and "scribble" in Figure 2 with "mask with false positives" and "mask with false negatives".
>
> ---
>
> **3. The impact on AMCP if DINO and CRF are replaced.**
>
> AMCP is based on the difference between the contents in the original and in/outpainted images. We leverage an image projector, i.e., DINO, to capture object semantic information to coarsely locate the object and utilize the CRF to capture pixel-level color information to better segment the object boundary.
>
> As shown in Tab A and Tab B, we conduct ablation studies to replace DINO with MoCov2 [11] and SwAV [7] and replace CRF [30] with the bilateral solver [2] respectively. We notice that the performance of AMCP only exhibits a slight variation when these replacements are applied.
>
> |Image Projector| MoCov2 [11] | SwAV [7] | DINO [8] |
> |:----:| :----: | :----: | :----: |
> | IoU | 80.1 | 79.7 | 80.6 |
>
> Table A: Ablation on image projector with coarse mask prompts on ECSSD.
>
> |Color potential|  None  | BS [2] | CRF [30]|
> |:----:| :----: | :----: | :----: |
> | IoU | 71.4 | 79.1 | 80.6 |
>
> Table B: Ablation on the method to calculate color potential with coarse mask prompts on ECSSD. BS: bilateral solver [2].
>
> ---
>
> **4. A fair comparison should be made with SAM. The comparison in the paper is not fair.**
>
> We report the performance of SAM (ViT-Base) as shown in Tab C while we think this comparison is not exactly fair as SAM is a supervised method with (1) the large-scale supervised training with labeled data, and (2) the large number of parameters. Different from the supervised pipeline employed by SAM, PaintSeg demonstrates that satisfactory segmentation can be achieved in a fully unsupervised manner by contrasting original and in/outpainted images. We believe the findings from PaintSeg remain valuable even if its performance is suppressed by SAM.
>
> For comparison in Tab 2 and Tab 3 in the paper, all methods are equipped with the same prompts (point or box) to ensure a fair comparison. Specifically, all compared point-prompt methods are designed to accept point prompts (following the conventional evaluation protocol). For box-prompt methods, we augment the compared methods by first cropping the object box region and conducting segmentation on the cropped images. In this way, we consider that those comparisons are fair and can reflect the effectiveness of PaintSeg.
>
> | Method | GrabCut (point) | Berkeley (point) | DAVIS (point) |  PASCAL VOC (box) | MVal (box) |
> |:----| :----: | :----: | :----: | :----: | :----: |
> | SAM [29] | 86.7 | 83.3 | 71.1 | 66.4 | 83.0 |
> | PaintSeg| 84.4 | 70.0 | 69.4 | 59.7 | 69.6 |
>
> Table C: Comparison with SAM for point and box prompted segmentation.
>
> ---
>
> **5. Pseudocode for more clear demonstration.**
>
> Thanks for your suggestion. We demonstrate the pseudocode of AMCP in Algo A (in the PDF attached in the general response). We will add the pseudocode in the revision.

---

### Author Rebuttal · Authors · 2023-08-08

We sincerely thank the reviewers for their valuable time and efforts.

We include the additional figure (Fig A) and pseudo code (Algo A) in the PDF file along with additional tables in the response for each question. We hope those responses could clarify your confusion, and we are more than happy to provide further explanations if needed.

---

### Decision · Program_Chairs · 2023-09-21

**Decision:**

Accept (poster)

**Comment:**

The work received favorable reviews. The reviewers found the idea of using alternating inpainting and outpainting steps to create segmentation masks without additional supervision novel and principled. The initial clarity concerns raised by the reviewers were well addressed by the rebuttal.

The AC agrees with the reviewers assessment - the authors propose a creative training-free approach for using generative models for segmentation. The AC requests the authors to incorporate the reviewers feedback and the rebuttal into the final version to further enhance the quality and impact of their work. The AC is pleased to recommend accepting!